# Comparative Analysis of Gene Expression in Fibroblastic Foci in Patients with Idiopathic Pulmonary Fibrosis and Pulmonary Sarcoidosis

**DOI:** 10.3390/cells11040664

**Published:** 2022-02-14

**Authors:** Jan C. Kamp, Lavinia Neubert, Helge Stark, Jan B. Hinrichs, Caja Boekhoff, Allison D. Seidel, Fabio Ius, Axel Haverich, Jens Gottlieb, Tobias Welte, Peter Braubach, Florian Laenger, Marius M. Hoeper, Mark P. Kuehnel, Danny D. Jonigk

**Affiliations:** 1Department of Respiratory Medicine, Hannover Medical School, 30625 Hannover, Germany; gottlieb.jens@mh-hannover.de (J.G.); welte.tobias@mh-hannover.de (T.W.); hoeper.marius@mh-hannover.de (M.M.H.); 2Biomedical Research in Endstage and Obstructive Lung Disease Hannover (BREATH), German Center for Lung Research (DZL), 30625 Hannover, Germany; neubert.lavinia@mh-hannover.de (L.N.); stark.helge@mh-hannover.de (H.S.); hinrichs.jan@mh-hannover.de (J.B.H.); ius.fabio@mh-hannover.de (F.I.); haverich.axel@mh-hannover.de (A.H.); braubach.peter@mh-hannover.de (P.B.); laenger.florian@mh-hannover.de (F.L.); kuehnel.mark@mh-hannover.de (M.P.K.); jonigk.danny@mh-hannover.de (D.D.J.); 3Institute for Pathology, Hannover Medical School, 30625 Hannover, Germany; boekhoff.caja@mh-hannover.de (C.B.); seidel.allison@mh-hannover.de (A.D.S.); 4Department of Diagnostic and Interventional Radiology, Hannover Medical School, 30625 Hannover, Germany; 5Department of Cardiothoracic, Transplant and Vascular Surgery, Hannover Medical School, 30625 Hannover, Germany

**Keywords:** fibroblastic foci, sarcoidosis, idiopathic pulmonary fibrosis, laser microdissection

## Abstract

Background: Fibroblastic foci (FF) are characteristic features of usual interstitial pneumonia (UIP)/idiopathic pulmonary fibrosis (IPF) and one cardinal feature thought to represent a key mechanism of pathogenesis. Hence, FF have a high impact on UIP/IPF diagnosis in current guidelines. However, although less frequent, these histomorphological hallmarks also occur in other fibrotic pulmonary diseases. Currently, there is therefore a gap in knowledge regarding the underlying molecular similarities and differences of FF in different disease entities. Methods: In this work, we analyzed the compartment-specific gene expression profiles of FF in IPF and sarcoidosis in order to elucidate similarities and differences as well as shared pathomechanisms. For this purpose, we used laser capture microdissection, mRNA and protein expression analysis. Biological pathway analysis was performed using two different gene expression databases. As control samples, we used healthy lung tissue that was donated but not used for lung transplantation. Results: Based on Holm Bonferroni corrected expression data, mRNA expression analysis revealed a significantly altered expression signature for 136 out of 760 genes compared to healthy controls while half of these showed a similar regulation in both groups. Immunostaining of selected markers from each group corroborated these results. However, when comparing all differentially expressed genes with the *fdr*-based expression data, only 2 of these genes were differentially expressed between sarcoidosis and IPF compared to controls, i.e., calcium transport protein 1 (*CAT1*) and SMAD specific E3 ubiquitin protein ligase 1 (*SMURF1*), both in the sarcoidosis group. Direct comparison of sarcoidosis and IPF did not show any differentially regulated genes independent from the statistical methodology. Biological pathway analysis revealed a number of fibrosis-related pathways pronounced in IPF without differences in the regulatory direction. Conclusions: These results demonstrate that FF of end-stage IPF and sarcoidosis lungs, although different in initiation, are similar in gene and protein expression, encouraging further studies on the use of antifibrotic agents in sarcoidosis.

## 1. Introduction

Idiopathic pulmonary fibrosis (IPF) is a devastating chronic pulmonary disease, characterized by aberrant deposition of extracellular matrix and resulting in progressive and irreversible pulmonary remodelling [1]. IPF typically affects people of middle or elevated age (mean age at diagnosis 66 years) [ATS Guidelines]. The diagnosis of IPF is primarily based on a typical radiological and/or histological injury pattern, the usual interstitial pneumonia (UIP) [2]. Histologically, UIP is characterized by a disturbed pulmonary architecture, a bronchiolization of the peripheral airways (i.e., aberrant alveoli with hyperplastic bronchiolar epithelial lining), and myogenic metaplasia [3]. In addition, one characteristic histological feature of UIP is the presence of fibroblastic foci (FF) which are found not solely, but also in early stages of disease development [3]. FF are composed of an accumulation of fibroblasts and/or myofibroblasts within a background of immature extracellular matrix and previous studies suggested an association of FF to disease genesis, progression and overall poor prognosis in UIP [4]. According to current guidelines of the American Thoracic Society (ATS), FF are one of the most important histopathological features for the diagnosis of UIP/IPF and their presence leads to an at least “probable” UIP pattern [2]. As IPF is a chronically progressive and devastating disease with a median transplant-free survival of about 3 years, an outcome comparable to that of solid organ malignancies; this diagnosis is associated with multifarious existential consequences for affected patients [5].

Although less frequent, FF can also manifest in a handful of other fibrosing pulmonary diseases like sarcoidosis [6], non-specific interstitial pneumonia (NSIP) [7], and chronic hypersensitivity pneumonitis [8].

Sarcoidosis is a granulomatous systemic disease predominantly affecting the lungs. The knowledge on its pathogenesis is still limited, but it is generally believed that at least one agent induces a self-maintaining immune reaction that results in a chronic granulomatous remodelling of the lungs, sometimes capable of inducing progressive pulmonary fibrosis [9]. The histopathological diagnosis of pulmonary sarcoidosis is based on the presence of coalescing, non-necrotizing granulomas, predominantly localized along the pleural lymphatic vessels, the interlobular septa, and the broncho-vascular bundles [10]. These granulomas are composed of epithelioid cells and multinucleated giant cells and surrounded by lymphocytic aggregates. In chronic pulmonary sarcoidosis, granuloma can undergo a fibrotic transition over time, resulting in an end-stage fibrotic lung disease, resembling a UIP pattern [11]. Typical radiological and histological hallmarks of IPF and sarcoidosis are presented in Figure 1.

To date, data on the prevalence of FF in variant fibrosing pulmonary diseases is very limited. In addition, it is still unclear if the molecular signaling pathways and pathomechanisms contributing to their formation are similar across different entities. However, based on the comparable histomorphology, it can be hypothesized that there is a related molecular background of FF in variant fibrosing lung diseases. To approach this issue, we selected lungs from clinically well-characterized IPF and sarcoidosis patients as well as healthy controls and performed a molecular comparison of FF from both entities in a compartment-specific comprehensive analysis.

## 2. Materials and Methods

### 2.1. Histopathology

The Institute of Pathology at Hannover Medical School archives a large number of lung specimens. Among these, there are about 1000 lung explants from patients who underwent lung transplantation for end-stage disease. Explants are consistently worked-up in a standardized manner immediately after explantation as described elsewhere [12] enabling high quality tissue analyses. While screening these lung explants, a high prevalence of FF was found in IPF lungs as expected, but also in sarcoidosis; therefore, these two entities were selected for further analyses. The identification of FF was performed using hematoxylin-eosin stained slices.

Our study groups comprised *n =* 33 subjects. Of these, *n* = 6 were diagnosed as IPF and *n =* 6 as pulmonary sarcoidosis, respectively. *n =* 21 control samples were obtained from healthy lungs that were donated, but not used for lung transplantation (so-called downsizing lungs, *n =* 21 different donors): sometimes, allocated lung transplants are oversized in regard to the recipients’ thorax. In such a situation, transplants are surgically downsized in order to prevent post-transplant oversize-related problems. However, the resected lung tissue is also healthy and vital and therefore suitable as control material. The histopathological diagnosis of sarcoidosis / IPF and the prevalence of FF were confirmed in all selected cases by two experienced pulmonary pathologists (LN, DDJ).

To date, there is no generally accepted histomorphological definition of FF. Based on our personal practical experience from > 10 years we specified the following criteria. In our experience, the obligatory criterion for the identification of FF is the accumulation of fibroblasts and/or myofibroblasts, perpendicularly oriented to the alveolar septae. Further defining criteria are divided into major and minor criteria. Major criteria include the alignment of myofibroblasts in a fascicular order, a background of immature extracellular matrix, and protrusion into the alveolar space. Minor criteria include covering by hyperplastic type-II pneumocytes or bronchiolar epithelial cells, existence at the forefront of fibrotic remodeling, directly adjacent to ventilated air spaces, and edema of the extracellular matrix. Lesions are designated as incomplete FF if at least 1 major and 1 minor criterion are missing [2,12,13].

### 2.2. Laser Capture Microdissection

Samples of both entities were obtained from *n =* 6 patients, respectively, using laser microdissection with the aim of generating compartment-specific RNA isolates. For this purpose, formalin-fixed paraffin-embedded (FFPE) lung tissue was prepared in a specific manner. First of all, sections with approximately 10 µm thickness were cut and placed on membrane coated RNAse free microscopy slides (mmi slides RNAse FREE, MMI Molecular Machines and Industries AG, Glattbrugg, Switzerland). Following this, sections were uncoated from paraffin using a descending alcohol series and then stained with filtered haemalum, but not with eosin to prevent RNA damage. All steps were performed in compliance with the usual hygiene regulations in order to prevent contaminations. 30 slides in 2 sections were used for laser dissection from each case. Laser microdissection was performed using the CellCut Plus system (MMI Molecular Machines and Industries AG, Glattbrugg, Switzerland). Selection and encircling of FF was conducted using Cell Tools V5.0 software (MMI Molecular Machines and Industries AG, Glattbrugg, Switzerland). Finally, encircled regions were laser-cut and withdrawn using isolation caps (MMI Molecular Machines and Industries AG, Glattbrugg, Switzerland). All samples from one patient were collected in one tube.

### 2.3. Gene Expression Analysis

Compartment-specific RNA was isolated from the obtained tissue extracts using the RNeasy FFPE Kit (Qiagen, Venlo, The Netherlands). RNA content was measured using the Qubit RNA IQ Assay (Thermo Fisher Scientific, Waltham, MA, USA) guaranteeing a minimum of 100 ng in each sample. Samples were then analyzed using a commercial panel on 760 fibrosis-specific genes (nCounter Human Fibrosis V2 Panel) and the nCounter Analysis System (NanoString Technologies, Seattle, WA, USA, respectively). Afterwards, normalization of counts was performed using the nSolver analysis software version 3.0 (NanoString Technologies, Seattle, WA, USA). Therefore, 10 internal reference genes were used, as predefined by the manufacturer. Well established housekeeping genes (glucuronidase beta (*GUSB*) and phosphoglycerate kinase 1 (*PGK1*) were designated as reference genes for standardization of measurements. Further analyses on the ascertained log2 mRNA counts were performed using R software version 3.4.4 (R Foundation for Statistical Computing, Vienna, Austria) and the nCounter Advanced Analysis module version 1.1.5. The absolute gene expression results of both entities were analyzed and compared with each other. A Shapiro-Wilks test performed on all intragroup gene expressions showed a predominant normal distribution of data. Hence, t-tests were used for pairwise comparisons and ANOVA for multi group comparisons. False discovery rates (*fdr*) were calculated and values < 0.05 were considered statistically significant. Statistical analysis was concluded by correction for multiple testing using the Holm Bonferroni method. Biological pathway analysis was performed using the GeneOntology database as well as gene-pathway associations supplied by the manufacturer.

### 2.4. Protein Expression Analysis

Paraffin sections (2 μm) of both entities were used for complementary immunostaining. Immunohistochemical targets were selected based on the disease-specific mRNA profile (see Figure 1) and delineated involved molecular pathways (see Table 1). Details about all utilized antibodies are depicted in Appendix A.

## 3. Results

### 3.1. Clinical Information

In this study, lung tissue samples from *n =* 6 IPF patients and *n =* 6 sarcoidosis patients were compared with each other as well as with *n =* 21 healthy controls. Sex distribution differed with 1 and 4 female patients in the IPF and sarcoidosis group, respectively, while age distribution was very similar in both groups. Healthy controls samples were age and gender matched. However, as the healthy control samples were obtained from tissue that was donated but not used for lung transplantation, clinical information cannot be provided in detail here due to data protection regulations and respecting the anonymity of donors. More details are shown in Appendix A.

### 3.2. Gene Expression

The mRNA expression analysis revealed a significantly altered expression signature for 375 out of 760 genes compared to controls as depicted in Appendix A. Of these, 264 showed similar regulation in FF from both disease entities. After Holm Bonferroni correction, 136 genes still showed a significantly altered expression compared to controls. Of these, 69 showed a similar expression in both groups, while 41 showed an altered expression solely in the sarcoidosis group and 26 solely in the IPF group, as depicted in Figure 2. While directly comparing FF from sarcoidosis and IPF, there were no differentially regulated genes. The 12 genes which showed the strongest, although not significant difference between both entities are depicted in Figure 3.

Based on the Holm Bonferroni corrected results, 67 of these 136 genes showed an altered expression compared to controls solely in sarcoidosis or in IPF. However, with regard to the *fdr*-values, only 2 of all these genes remained differentially expressed as depicted in Appendix A, i.e., decreased expression of calcium transport protein 1 (*CAT1*) and increased expression of SMAD specific E3 ubiquitin protein ligase 1 (*SMURF1*), both in the sarcoidosis group. Raw data are deposited in Appendix A.

### 3.3. Biological Pathway Analysis

In comparison of FF from both entities with healthy control lungs, a number of biological pathways showed increased activity in FF from both entities as shown in Table 1. In FF from sarcoidosis, these were only tendencies. While comparing FF from sarcoidosis and IPF with each other, IPF showed a distinct up-regulation of a number of fibrosis-related pathways and a down-regulation of certain inflammatory pathways.

### 3.4. Protein Expression

Based on the disease specific mRNA expression signatures, we selected fibroblast activation protein alpha (FAP), vascular cell adhesion molecule 1 (VCAM1), cluster of differentiation 86 (CD86) as well as collagen types 5 (COL5) and 6 (COL6) as appropriate markers for immunostaining. In addition, we used marker of proliferation Ki-67 (Ki-67) as a proliferation marker. As depicted in Figure 4, FAP and VCAM1 showed a similar expression pattern in both entities. As shown in Figure 5, CD86 and COL5 showed a similar expression in both IPF and sarcoidosis lungs while COL6 was enhanced in IPF lungs. Ki-67 showed no enhancement in fibroblastic foci from both disease entities. These results are in line with the mRNA expression profile.

## 4. Discussion

IPF and sarcoidosis represent chronic pulmonary diseases with various degrees of progressive pulmonary fibrosis. In end-stage disease of both entities, bilateral lung transplantation remains the only viable treatment option. Although histopathological findings differ between IPF and sarcoidosis, FF can be present in both entities. However, usually FF are larger and more prevalent in IPF, than in sarcoidosis [6] or other fibrotic lung diseases [4]. To date, it is still unclear which molecular mechanisms contribute to their evolution and in particular if these mechanisms – and FF, for that matter - are the same in different entities. In this work, compartment-specific analysis of FF was performed in end-stage organs from IPF and sarcoidosis patients who underwent lung transplantation.

One fundamental finding of our study was the striking similarity of differentially regulated genes when directly comparing IPF and sarcoidosis lungs, distinctly suggesting a common nature of FF, independent from the underlying disease. Even when comparing sarcoidosis and IPF lungs with healthy controls, 69 of the 136 genes with an altered expression profile showed a similar expression in both entities and only 2 genes showed a contrary-regulated expression. With regard to the significantly regulated biological pathways, regulatory patterns were basically the same in both entities, while particularly the fibrogenesis-related pathways were pronounced in IPF lungs. The fact that particularly inflammation-related pathways such as interferon type I/II, tumor necrosis factor production, toll-like receptor, chemokine, and T-cell receptor signaling were less pronounced in tissue from FF compared to healthy controls, might be a consequence of the compartment-specific tissue analysis and the comparison to mRNA from whole lung slices consisting of many other cell-types such as myocytes, endothelial and immune cells.

At first glance, there was a number of single genes with an altered expression signature between sarcoidosis and IPF lungs, as compared to controls, respectively. Here we focused on selected genes with an established relationship to fibrogenesis, sarcoidosis, and/or IPF.

In a former study by Guillotin et al., myofibroblasts within fibroblastic foci from IPF lungs had been isolated using laser microdissection and specifically analyzed [14]. Among others, clusters of genes associated with cell-cycle and cell adhesion were identified to be involved in the fibrogenesis in IPF. In our study, there was also an increased expression of several genes encoding for cell adhesion molecules or the cell cycle in both groups, e.g., *VCAM1* and integrin subunit beta 3 (*ITGB3*) [15] in sarcoidosis, neural cell adhesion molecule 1 (*NCAM1*) in both entities as well as anaphase promoting complex subunit 10 (*ANAPC10*) and proteasome 26S subunit, non-ATPase 13 (*PSMD13*) in IPF lungs. Hence, cell cycle and cell adhesion molecules appear to play an important role in the development of FF in both entities, supporting the hypothesis of a common nature of FF in variant disease entities.

Lysyl oxidase-like (LOXL) protein coding genes belong to a family of ECM crosslinking enzymes and particularly *LOXL1* and *LOXL2* are known to be strongly involved in the process of pulmonary fibrogenesis [16]. Increased *LOXL1* and *LOXL2* levels both on the mRNA and on the protein level have been described in IPF [17] and LOXL1 protein deficiency has been shown to prevent pulmonary fibrosis induced by transforming growth factor beta 1 (*TGF-B1*) overexpression in knockout mice by blocking fibrillar collagen organization and subsequent tissue stiffening [18]. In addition, *LOXL1* and *LOXL2* have been suggested as potential therapeutic targets in IPF [17]. In our study, *LOXL1* and *LOXL4* showed increased expression in the IPF group, as well as *LOXL2* and *LOXL4* in the sarcoidosis group. Thus, although not identical, there are distinct similarities with regard to the expression of LOXL proteins in FF from sarcoidosis and IPF lungs.

Matrix metalloproteinases (MMP) represent a family of proteins involved in the turnover of extracellular matrix in physiological and pathological processes, such as embryonic organogenesis and fibrotic tissue remodeling. Of these, *MMP7* is known to play a pivotal role in the pathogenesis of IPF [19,20]. In addition, increased *MMP1* expression was related to IPF associated with non-small cell lung cancer in a recent work [21]. However, data on MMP1 and MMP7 as potential serum biomarkers for IPF and/or sarcoidosis are controversial. Some groups suggested both markers as IPF-specific predictors [22,23] while another group reported higher serum and bronchoalveolar lavage fluid concentrations of MMP1 in sarcoidosis compared with higher concentrations of MMP7 in IPF in each fluid [24]. One study also demonstrated an increased mRNA expression of *MMP7* in FF from UIP patients within a compartment-specific analysis [25] and one further group found increased plasma concentrations of MMP7 in sarcoidosis patients compared to healthy controls [26]. In the present study, *MMP7* showed an increased expression in FF not only from IPF but also from sarcoidosis lungs. Additionally, *MMP1* showed increased expression in sarcoidosis. These results demonstrate again the molecular conformity of FF from both entities.

Basically, tyrosine kinases can be divided into receptor and non-receptor tyrosine kinases depending on their location in cells. Receptor tyrosine kinases (RTK) have been shown to contribute substantially to the fibrogenesis in IPF [27,28]. On that basis, the multi-kinase inhibitor Nintedanib has been approved for the treatment of IPF [29,30]. RTK targeted by Nintendanib are platelet-derived growth factor receptor (*PDGFR*), fibroblast growth factor receptor (*FGR*) and vascular endothelial growth factor receptor (*VEGFR*) [31].

The RTK MET proto-oncogene (*MET*) serves as a key regulator of invasive growth. In relation to fibrogenesis in IPF, *MET* regulates the proliferation and survival of fibroblasts as well as their migration across the ECM, making it a potential molecular target for antifibrotic therapies [27]. Phospholipase C Gamma 1 (*PLCG1*) mediates the production of two second messenger molecules involved in the regulation of intracellular signaling cascades [32]. Of note, *PLCG1* becomes activated in response to ligand-mediated activation of several RTK, including platelet-derived growth factor receptor (*PDGFR*) and fibroblast growth factor receptor (*FGFR*) [33]. In our study, there was an increased expression of *MET* and *PLCG1* in FF from IPF lungs within the IPF group, reinforcing the important role of RTK for the fibrogenesis in IPF.

SRC proto-oncogenes (*SRC*) subsume a number of non-receptor tyrosine kinases pivotal for several signal transduction and regulatory processes, as well as for cell proliferation and differentiation [34]. Phosphorylation and activation of these kinases is promoted among others by *PDGFR* [35]. SRC are involved in various molecular pathways such as epithelial to mesenchymal transition and myofibroblast differentiation, particularly in response to inflammatory stimuli. In addition, in vitro and in vivo experiments in which inhibitory effects on pulmonary fibrogenesis were observed using specific SRC kinase inhibitors [36,37] corroborate the crucial role of SRC for the development of variant fibrotic lung diseases. In line with these previous studies, SRC showed a higher expression in sarcoidosis lungs in our study.

Taken together, these results indicate the molecular similarity of FF in sarcoidosis and IPF suggesting that the pathogenic mechanisms behind FF are independent from the underlying disease: a FF is a FF, regardless of the underlying disease. This in turn encourages the use of antifibrotic agents approved for IPF also in progressive fibrosing sarcoidosis. Moreover, in the light of the appearance of FF in other fibrotic lung disease entities, the impact of their presence on diagnosis should be re-evaluated in current guidelines. To draw a comparison, in a former work we could show that bronchiolitis obliterans (BO) and alveolar fibroelastosis (AFE) are based on the same molecular processes independent from the respective disease entities [38]. Interestingly, by analogy here we see again a divergence of histomorphological and clinical presentations emphasizing the importance of multidisciplinary boards for interstitial lung diseases. Limitations of this study are the small sample size and the monocentric design. Future studies are needed to explore the portability to other fibrotic lung disorders.

## Figures and Tables

**Figure 1 cells-11-00664-f001:**
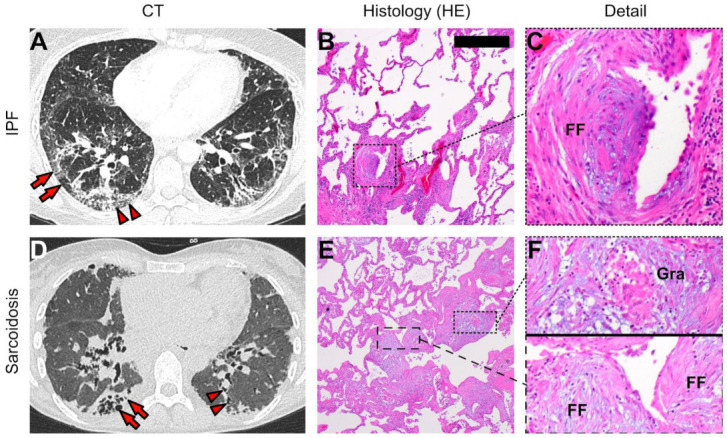
Radiological and histopathological features of sarcoidosis and IPF. (**A**) Exemplary axial high resolution computed tomography (CT) slide showing typical radiological predominantly subpleural distributed features of idiopathic pulmonary fibrosis (IPF) including septal thickening, mixed ground glass opacities, reticular densities (arrowheads), and areas of subpleural honeycombing (arrows); (**B**) typical histopathological features of IPF including disturbed pulmonary architecture (i.e., aberrant alveoli with hyperplastic bronchiolar epithelial lining, extensive myogenic metaplasia, and multiple fibroblastic foci (FF) magnified in (**C**); (**D**) exemplary axial high resolution CT slide showing typical hallmarks of fibrotic pulmonary sarcoidosis including reticular and cystic changes, traction bronchiectasis (arrowheads), and subpleural bullous changes (arrows); (**E**) histopathological hallmarks of sarcoidosis including focal organizing pneumonia and interstitial pneumonitis as well as the in (**F**) magnified multiple granuloma (Gra) with central giant cells and surrounding epithelioid cells and FF; HE, hematoxylin-eosin stained; scale bar equals 500 µm.

**Figure 2 cells-11-00664-f002:**
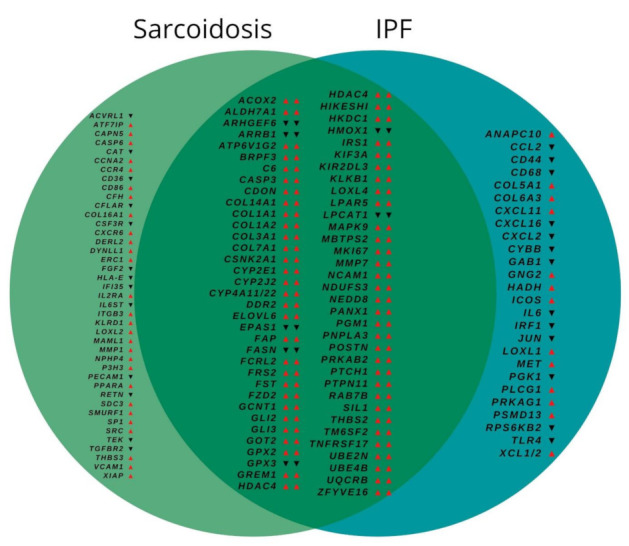
Venn diagram of differentially regulated genes. Arrows indicate significantly increased (arrow up) or significantly decreased (arrow down) expression of the respective genes compared to controls in fibroblastic foci from both disease entities compared to healthy control lungs. Holm Bonferroni corrected *p*-values < 0.05 were considered as statistically significant. In the overlapping area (middle), left and right arrow indicate the expression in sarcoidosis and IPF, respectively. IPF, idiopathic pulmonary fibrosis.

**Figure 3 cells-11-00664-f003:**
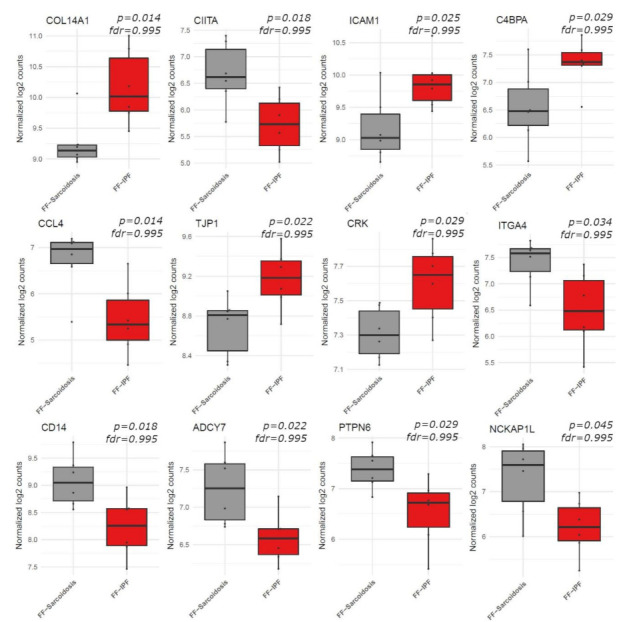
Gene expression—sarcoidosis vs. IPF. This figure shows the top 12 regulated genes between sarcoidosis and idiopathic pulmonary fibrosis (IPF) in box-plots. FF, fibroblastic foci; IPF, idiopathic pulmonary fibrosis; *fdr*, false discovery rate.

**Figure 4 cells-11-00664-f004:**
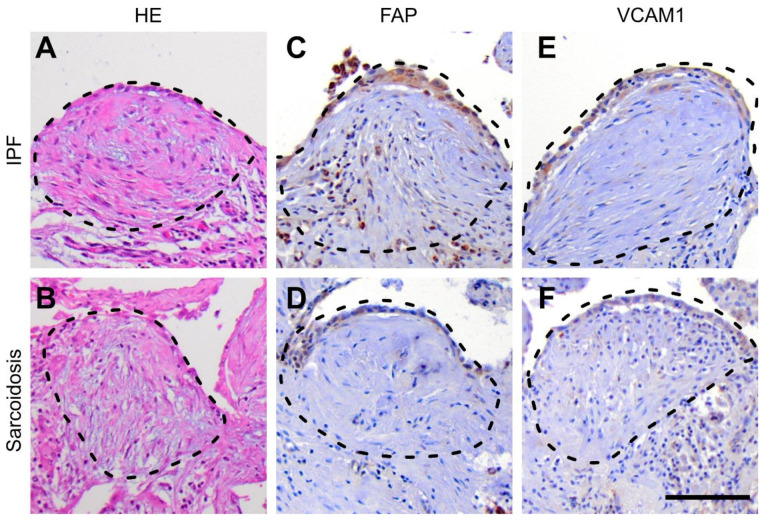
Immunohistochemical staining of fibroblastic foci. Histological morphology of fibroblastic foci (FF), hematoxylin-eosin (HE) stained, in (**A**) idiopathic pulmonary fibrosis (IPF) and (**B**) sarcoidosis showing characteristic myxoid arranged fibroblasts and myofibroblasts, a background of immature extracellular matrix, protrusion into the alveolar space, and a covering epithelial layer of hypertrophied type II pneumocytes; fibroblast activation protein alpha (FAP) showed a coherent staining pattern in FF from sarcoidosis (**C**) and IPF (**D**) lungs with a cytoplasmic enhancement of the covering epithelial layer and a low intensity cytoplasmic staining of scattered myofibroblasts (less pronounced in sarcoidosis); (**E**,**F**) vascular cell adhesion molecule 1 (VCAM1) showed a specific cytoplasmic enhancement in respiratory epithelial cells and less intensive also in myofibroblasts / connective tissue components (slightly pronounced in sarcoidosis; scale-bar equals 100 µm.

**Figure 5 cells-11-00664-f005:**
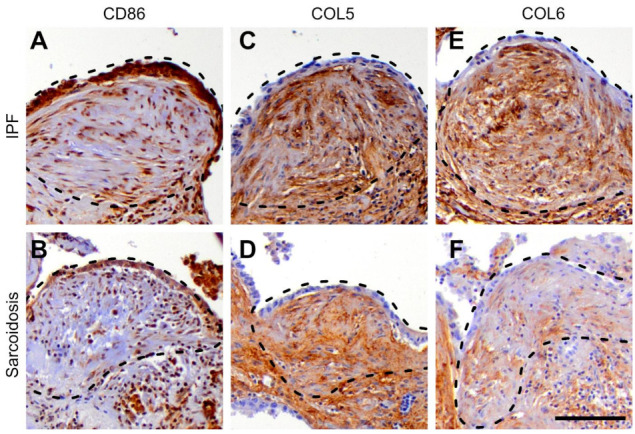
Immunohistochemical staining of fibroblastic foci. (**A**,**B**) Cluster of differentiation (CD86) showed a specific cytoplasmic enhancement in respiratory epithelial cells and also in myofibroblasts similar in both entities; (**C**,**D**) collagen type 5 (COL5) showed a similar staining of fibroblastic foci from both entities predominantly in myofibroblasts and the surrounding connective tissue; (**E**,**F**) Collagen type 6 (COL6) was also enhanced in myofibroblasts and connective tissue structures but markedly stronger in idiopathic pulmonary fibrosis (IPF) lungs than in sarcoidosis; scale bar equals 100 µm.

**Table 1 cells-11-00664-t001:** Changes in the expression of genes involved in several biological pathways.

Signaling Pathway	FF-S vs. Controls		FF-IPF vs. Controls		FF-IPF vs. FF-S	
	Regulation	*fdr*	Regulation	*fdr*	Regulation	*fdr*
Cell adhesion ^1^	Up	0.782	Up	0.437	Up	0.018
ECM structure ^1^	Up	0.165	Up	0.063	Up	0.018
ECM receptor interaction ^1^	Up	0.623	Up	0.437	Up	0.018
ECM organization ^2^	Up	0.83	Up	0.503	Up	0.005
Organization of collagen fibrils within ECM ^2^	Up	0.261	Up	0.036	Up	0.095
Innate immune system ^1^	Down	0.165	Down	0.011	Down	0.0001
Type I interferon signaling ^1^	Down	0.165	Down	0.031	Down	0.002
Type II interferon signaling ^1^	Down	0.343	Down	0.206	Down	0.009
MHC class I antigen presentation ^1^	Down	0.13	Down	0.036	Down	0.002
Activation of TNF production ^2^	Down	0.916	Down	0.038	Down	0.005
Activation of ERK1/ERK2 cascade ^2^	Down	0.14	Down	0.036	Down	0.884
NLR signaling ^2^	Down	0.276	Down	0.036	Down	0.241
TLR signaling ^1^	Down	0.926	Down	0.202	Down	0.001
T-cell receptor signaling ^1^	Down	0.926	Down	0.96	Down	0.013
Chemokine signaling ^1^	Down	0.957	Down	0.769	Down	0.018
Macrophages ^1^	Down	0.787	Down	0.091	Down	0.015

^1^ based on gene-pathway associations supplied by Nanostring; ^2^ based on the GeneOntology database; significant *fdr* values are written in bold letters; FF-IPF/FF-S, fibroblastic foci in IPF/sarcoidosis; IPF, idiopathic pulmonary fibrosis; *fdr*, false discovery rate; MHC, major histocompatibility complex; NLR, Nod-like receptor; HIF1A, hypoxia inducible factor; ERK1/2, extracellular signal-related kinase 1/2; ECM, extracellular matrix; TNF, tumor necrosis factor; TLR, toll-like receptor.

## Data Availability

Data is contained within the Appendix A. The data presented in this study are available in Appendix A.

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
