# Peer review of "Comparative Analysis of Gene Expression in Fibroblastic Foci in Patients with Idiopathic Pulmonary Fibrosis and Pulmonary Sarcoidosis"

_cells, 2022, doi:10.3390/cells11040664_

Round 1
Reviewer 1 Report
In this study the authors compared the expression of several genes in fibroblastic foci from IPF and pulmonary sarcoidosis patients. The study is original but there are several issues need to be clarified before publication.
- The title is very unclear. It should be changed to something like: Comparative analysis of gene expression in fibroblastic foci in patients with idiopathic pulmonary fibrosis and pulmonary sarcoidosis.
- Clarify whether the graphs presented in Figure 1 are original from the authors' study or previously reported by previous reports.
- The authors addressed "the current-problem-to-solve" in the following sentence in the introduction section "To date, it is still unknown how prevalently FF are in variant fibrosing pulmonary disease entities and if the respective molecular signaling pathways and pathomechanisms that contribute to their formation are the same" However, this is very unclear. The authors should state much clearer, first, their "working hypothesis" and second "how they are going to demonstrate their hypothesis" in the introduction section. This lack of clarity could be the "English language." Therefore, consultation with and correction by a native English speaker would also be very useful to improve the statements in the introduction section.
- In the method section, it is unclear how many patients were included in the study. These details should also be included in the method section. The reviewer can see that the details under the result section. These data should be transferred to the method section.
- The healthy controls should also be described in detail.
- There were 33 patients, but only 6 with sarcoidosis and 6 with IPF were described. Were the other patients diagnosed with a different disease? The authors should cite the reference to diagnose UIP/IPF and lung sarcoidosis. Were the 33 patients consecutive patients referred or consulting at the institution? What treatments were the patients receiving?
- Figure 2 and Figure B1: Do the up- or down-directed arrows indicate significant expression? Is it based on fdr? The abbreviation fdr should be defined in the legend. The authors wrote "activity," but I do not think data show the activity of the genes. These should be clarified. The positions of the figures' titles and the legends are also wrong. They should be together.
- Table 2. "increased activity" expression is wrong. Is it activity or expression? The title described only "increased" expression, but there are also "decreased" genes. They should write something like "Changes in the expression of genes involved in several biological pathways" to make it easier to follow.
- Figure 3. The figures should show the p values in each graph even though they are not significant.
- On page 193, 3.3 section: Functional analysis: What functional analyses were done? I understand only gene expression analysis was done. Did the authors study the function of each gene involved in each biological pathway? This is not easy to understand.
- In Figure 4, the title is separated from the legend. The title is not clear. The scale bar is excessively thick.
- In the discussion section, the authors again used the word "activity" it should be gene expression. The discussion is too long and unfocused. The authors should describe the literature and then describe what they found related to what is already known from the literature.
- The English language needs to be checked by a native speaker.
Reviewer 2 Report
This study compared gene expression profiles in fibroblastic foci from IPF and sarcoid lung explants using a commercial panel of 760 fibrosis specific genes. The authors conclude that the gene expression profile is similar between these two entities, raising the possibility that anti-fibrotics useful in IPF may have a role in fibrotic sarcoid.
Major Comments:
- The presentation of the data is confusing. I may be missing something regarding the comparison of IPF vs. sarcoid. In the abstract, it is stated that 114 genes were differentially expressed in the disease groups relative to controls. In the results section, 67 genes are differentially expressed (when I count in figure 2, this is 69) in both disease entities relative to controls, plus 41 in sarcoid and 26 in IPF. No mention is made of the only 2 genes that were differentially expressed in the results section. This is again alluded to in discussion.
Minor Comments:
- If clinical data on the subjects is to be presented, more detailed information regarding pulmonary function and radiographic appearance would be helpful in addition to simply age and gender that is taking up space in table 1.
- It would be useful to show magnitude of change in gene expression for the genes presented. There is a supplemental table C1 that was not included for review.
- Figure 3 should show the fdr for the selected genes.
Round 2
Reviewer 1 Report
Many parts of the manuscript are still confusing.
The authors did not follow my suggestion to ask someone fluent in English or a native English speaker to check all the manuscript.
Line 29: “Functional analysis” should be changed. The authors did not study function of the genes.
Lines 41 to 44: The conclusion is overstated: “These results suggest that the molecular mechanisms behind the development of FF are similar in sarcoidosis and IPF, suggesting a re-evaluation of current guidelines and encouraging further studies on the use of antifibrotic agents in sarcoidosis”
Why the authors believe because there are similar expressions of genes there should be similar “molecular mechanisms”, and why this would require “re-evaluation of current guidelines” of therapy.
Line 130: The reviewer does not understand yet the subjects of the study.
“Our study groups comprised n=33 patients. Of these, n=6 were diagnosed as IPF and n=6 pulmonary sarcoidosis, respectively. N=21 control samples…”
In this sentence the authors wrote 33 patients, where are the 21 remaining patients. Are these 21 patients healthy controls? In that case they should write “Our study groups comprised n=33 subjects”.
Under the Discussion section:
Line 305: “Although histopathological aspects differ between IPF and sarcoidosis, FF can present in both entities.” In this sentence “aspects” should be findings and “FF can present in both entities” should be “FF can be present in both entities”
How can the reviewer believe that the authors asked a native speaker to check the entire manuscript.
In line 307 again : “more prevalent in IPF, then in sarcoidosis [6] or other fibrotic lung diseases” . In this sentence “then in sarcoidosis” should be “than in sarcpoidosis”.
There are many of these mistakes that made the manuscript very difficult to read.
Please give the manuscript to a Senior scientist to make it more readable.
Reviewer 2 Report
Eliminate table one. This information can be summarized in text.Author Response
Please see the attachment.
